# Effect of academic stress, educational environment on academic performance & quality of life of medical & dental students; gauging the understanding of health care professionals on factors affecting stress: A mixed method study

**Muhammad Hassan Wahid**[1], **Mifrah Rauf Sethi**[2], **Neelofar Shaheen**[3]*, **Kashif Javed**[1], **Ijlal Aslam Qazi**[1], **Muhammad Osama**[1], **Abdul Ilah**[1], **Tariq Firdos**[1]

1 Undergraduate Medical Student, Peshawar Medical College, Riphah International University, Islamabad, Pakistan, 2 Department of Mental Health, Psychiatry and Behavioral Sciences, Peshawar Medical College, Riphah International University, Islamabad, Pakistan, 3 Department of Health Profession Education and Research, Peshawar Medical College, Riphah International University, Islamabad, Pakistan

* neeloferadeel@gmail.com

## Abstract

### Introduction

Throughout their academic careers, medical and dental students face challenges that cause varying levels of stress, affecting their academic performance and quality of life (QoL). Our study aims to ascertain the effect of academic stress and the educational environment on the QoL and academic performance of medical and dental students, encompassing the perspectives of both students and healthcare professionals.

### Methods

A mixed-method research was conducted from February to May 2022, comprising students from a medical and dental college in Pakistan. During Phase 1, the students participated in the cross-sectional survey and completed the WHO Quality of Life Scale (WHOQOL-BREF), Academic Stress Scale, and Dundee Ready Educational Environment Measure (DREEM) Inventory questionnaires. Academic performance was evaluated through last year's annual assessment results of the students. During Phase 2 of the study, interviews with healthcare professionals who had experience as the students' counsellors were conducted.

### Results

The mean age of the sample (n = 440) was 22.24 ±1.4 years. The Cronbach Alpha reliability of the DREEM inventory was 0.877, that of the Academic Stress Scale was 0.939 and the WHOQOL scale was 0.895. More than half of the students (n = 230, 52.3%) reported better

**Data Availability Statement:** The SPSS data coded zip file is attached to this submission. Some data cannot be shared publicly because of confidential academic records of the participants. Details of qualitative data and academic records are available from the PRIME Foundation Institutional Ethics Committee (contact via Dr Hala Rajab hala.rajab@prime.edu.pk, halarajab202@gmail.com) for researchers who meet the criteria for access to confidential data. The mentioned contact person is the secretary Institutional Ethics Committee.

**Funding:** The author(s) received no specific funding for this work.

**Competing interests:** The authors have declared that no competing interests exist.

**Abbreviations:** MHW, Muhammad Hassan Wahid; MRS, Mifrah Rauf Sethi; NFS, Neelofar Shaheen; KJ, Kashif Javed; IAQ, Ijlal Aslam Qazi; MO, Muhammad Osama; AI, Abdul Ilah and; TF, Tariq Firdos.

QoL and the majority perceived a positive educational environment (n = 323, 73.4%) and higher academic stress (n = 225, 51.1%). Males had significantly more academic stress (p<0.05). Those who perceived a positive educational environment and better QoL had better academic performance (p<0.05). Academic performance was positively and significantly correlated with QoL and academic stress (p = 0.000). In qualitative analysis, 112 codes were generated which converged into 5 themes: challenging educational environment, psychological need and support, individual differences, relationship and family life, and adjustment issues.

## Conclusion

Medical and dental students encounter a myriad of challenges, along with significant academic stress, which detrimentally affects their academic performance, despite perceiving a positive educational environment. Conversely, a better QoL is associated with improved academic performance.

## Introduction

Academic stress is defined as a body's response to academic-related demands that exceed the adaptive capability of students [1]. According to a survey carried out by the team of "Dictionary of Human Geography" in the U.S.A, academic stress is more often seen in medical and dental students as the total prevalence of stress in medical and dental students was reported as 63% and the prevalence of severe stress was 25% which affected their quality of life (QoL) and academic performance directly [2]. High academic stress among medical and dental students is a phenomenon that transcends socio-cultural factors, economic status, and course patterns [3]. The higher demands associated with academic studies may interfere with demands in other domains of life and vice versa [4]. Mental health is especially affected by perceived stress among medical and dental students from aged-matched non-medical student peers [5]. Medical and dental students experience high cynicism and high emotional exhaustion due to work overload [5].

Academic self-efficacy, perceived stress and preferred learning styles have been linked to academic performance and significantly affect medical and dental college students [6]. Perceived academic stress affects one's QoL. QoL is an overarching term for the quality of various domains in life, therefore, it is a subjective multidimensional concept that defines a standard level for emotional, physical, mental, and social well-being. It is believed that one must possess a good quality of life for one's proper efficiency in work and good mental and physical status [7]. Students who disclose utilizing a multitude of self-care practices throughout their undergraduate journey in medical and dental colleges reported a decrease in the strength of the relationship between perceived stress and QoL [8].

Medical schools can produce intense psychological distress in their students. However, physical exercise, sufficient leisure time and a good lifestyle can help students to be potential leaders in the field of health [9]. QoL is an essential component of learning and has strong links to the practice and study of medicine and dentistry. There is burgeoning research to suggest that medical students are experiencing mental and psychological troubles such as perceived academic stress and burnout [10]. Acute academic stresses are positive and enable a person to perform better from the start of the career, but the chronic form is the silent killer.

No one is shielded from stress, but it is more prevalent in medical and dental students. The major contributing factors are an unbalanced lifestyle and study overload [11].

Mental and psychological QoL scores were lower among medical students than among the age-matched general population. The prevalence of stress symptoms was similar regardless of race, colour, and cast [12]. The students had to struggle hard to maintain their academic performance and had been having little physical activity and disturbed sleep patterns [12]. Perception of the educational environment of the institution was more positive than negative among medical students. Better performance in examinations was associated with better academic self-perception and social self-perception in students [13]. The association between the educational environment and QoL is complex and multifaceted, and research reveals that a positive educational environment serves as a moderator to QoL of medical students [14]. A maladaptive response of students to different stimuli in a complex environment of medical schools can lead to poor QoL [14].

Drawing upon this body of evidence, our study aims to ascertain the effect of academic stress and the educational environment on the QoL and academic performance of medical and dental students, encompassing the perspectives of both students and healthcare professionals, who counselled students to overcome obstacles while pursuing their academic goals.

## Methodology

This sequential-explanatory mixed-method study was conducted on medical and dental students of two private institutions in Peshawar, Pakistan from February 2022 to May 2022. The participants were recruited in March 2022. The age range was between 18 to 25 years. All the students above the age of 18 years were included in the study. The study excluded students with mental health conditions and first-year medical students who had recently started their medical studies and their professional examination results could not be acquired. The authors had access to the demographic data of the participants and individual participants could be identified during or after the data collection.

During Phase 1 (Quantitative part) structured, reliable and validated questionnaires were distributed among the medical and dental students. In phase 2 (Qualitative part) interviews were conducted with healthcare professionals to explore their perspectives on the factors they considered relevant to the student's academic stress.

### Phase 1

A cross-sectional survey was conducted including all medical and dental students from $2^{nd}$ to final year, using the purposive sampling technique. Ethical approval was taken from the Ethical Review Committee of PRIME Foundation (PRIME/ERC/2022-01), Pakistan, and permission was obtained from the concerned authorities (Vice-chancellor/Dean) of the institutions before the start of data collection. Every student involved in the study was provided with a clear understanding of the study's goals, and their participation was contingent upon granting written informed consent. Prior to data collection, demographic information was gathered through structured questionnaires. Subsequently, the data collection instruments were administered to the participants.

**World Health Organization, Quality of Life scale: (WHOQOL-BREF).** The World Health Organization Quality of Life (WHOQOL-BREF) was used to assess the QoL, which is an empirical instrument to assess QoL [15]. It comprises 26 items and measures the domains of physical health; psychological health; social relationships; and environment in addition to measuring general QoL. Responses to questions are on 5 points Likert scale where "1" represents "disagree" or "not at all" and "5" represents "completely or extremely agree".

Cronbach's alpha was 0.85, 0.83, 0.62, and 0.81, respectively, for the physical, psychological, social, and environmental domains, and 0.92 for the total scale. Previous studies employing Confirmatory Factor Analysis on the WHOQOL-BREF have documented good to excellent reliability properties and better performance on initial validity tests [15]. Higher scores based on the mean average indicate better quality of life and vice versa [15]. Those who scored more than the mean score of 61.97 were considered to have better QoL and those less than 61 were categorized as having poor QoL.

**Academic performances.** In our study, annual professional exam results were taken as a measure of academic performance. The grading scale used was 1. (10–35%), 2 (35–60%), 3. (60–85%), and 4. Very high (85–100%).

**Dundee Ready Educational Environment Measure (DREEM) inventory.** The 50-item Dundee Ready Education Environment Measure (DREEM) questionnaire assesses the quality of the educational environment, and responses are rated on a five-point Likert scale, spanning from 0, representing strong disagreement, to 4, signifying strong agreement. The comprehensive cumulative score encompasses values from 0 to 200. Within the 50-item scale, 9 items are negatively stated and necessitate subsequent reverse scoring. The evaluation of students' environmental perception was stratified into the following categories: an aggregate score between 0 and 50 indicated a very poor perception of the environment, a range of 51 to 100 indicated plenty of problems, the interval of 101 to 150 signifies a preponderance of positive aspects relative to negative facets, and a score falling within 151 to 200 was characterised as indicative of an excellent environment. The questionnaires have been divided into five subscales: students' perception of learning (SPL), students' perception of teachers (SPT), students' academic self-perceptions (SAP), and student's perception of the atmosphere (SPA), and students' social self-perception. For the DREEM inventory, the cut-off scores are determined by higher than mean average and categorized as excellent environment, more positive than negative environment, and plenty of problems. The details of the scoring procedure are described by McAleer and Roff for interpreting the overall and subscale scores [16, 17]. DREEM is a widely accepted and universally validated tool, demonstrating an internal consistency of 0.86 and a test-retest reliability of 0.595 (p < 0.001). Previous studies utilizing Confirmatory Factor Analysis have affirmed the good model fit for the five-factor structure of DREEM-50 [16, 17].

**Academic stress scale.** The 40-items academic stress scale was originally developed by Kim (1970) [18]. It is a five point Likert scale ranging from "no stress i.e., 0" to "extreme stress i.e., 4". The scale was classified into five sub-areas containing 8 items each: personal inadequacy, fear of failure, interpersonal difficulties with teachers, teacher-pupil relationship/teaching methods and inadequate study facilities. The Academic Stress Scale has a good internal consistency score with a Cronbach alpha of 0.70 [18]. The total number of items was 40. Therefore 160 is the maximum possible score and the highest score for each factor would be 32 [18]. Those who scored more than the mean score of 67.13 were considered to have high academic stress and those less than the mean score of 67 had no academic stress. Overall, the higher scores on the mean average indicated more academic stress.

## Phase 2

Phase 1 was followed by in-depth qualitative interviews with healthcare professionals from different departments of medical sciences. Healthcare professionals were the faculty members who were purposively selected as they had more than 3 years of experience in counselling, advising, or mentoring students to cope with academic stress challenges. Faculty members who fulfilled the eligibility criteria were approached to consent to one-to-one interviews. The principal investigator conducted in-depth interviews with healthcare professionals in order to

comprehensively investigate their viewpoints concerning the determinants contributing to academic stress among students. Each interview session lasted between 40 to 60 minutes, with the interviews continuing until the point of data saturation, whereby no novel insights were forthcoming. The saturation point was attained upon the completion of 10 interviews. The verbatim content of the audio-recorded interviews was transcribed and subsequently subjected to validation from the interviewees. The interview guide was developed by following the protocol for questionnaire development [19] and was validated by the experts.

### Data analysis

**Phase 1: Quantitative phase.**    The analysis of the data was carried out using SPSS version 25. Basic variables were analysed using descriptive statistics for finding frequencies and percentages, means, and standard deviation. The internal consistency of the scales was measured through Cronbach's Alpha reliability and a value equal to or greater than 0.70 was considered satisfactory [20]. The Chi-square test was used to find the association between demographic variables, academic stress, academic performance, QoL and educational environment. In Addition, the Spearman correlation test was used to check the correlation between all four components (academic stress, QoL, academic performance and educational environment). The results of all the tests of significance were considered significant at $p < 0.05$. To assess the degree of association, the designated values of (rs) were categorized as follows: 0–0.19 indicating a weak correlation, 0.40–0.59 indicating a moderate correlation, 0.6–0.79 indicating a strong correlation, and 0.8–1 indicating a very strong correlation [21].

**Phase 2: Qualitative phase.**    For the qualitative component, we used thematic analysis using Virginia Braun and Victoria Clarke's six steps approach to generate themes from interview transcripts [22]. The six steps included familiarization with the data, generating initial codes, searching for themes followed by reviewing, understanding, naming the themes, and finally producing the results [22]. To ensure the quality of qualitative data we applied triangulation and member-checking methods.

## Results

Our study involved 500 students from a medical and a dental college with a participation rate of 80% (n = 440). The average age of the participants was 22.24±1.5 years, with an age range of 18 to 25 years. Of the participants, 51% were male (n = 224), and 62.3% (n = 274) indicated they lived in a nuclear family setup. The internal consistency of the instruments was strong, with a Cronbach's alpha reliability coefficient of 0.877 for the DREEM inventory, 0.939 for the Academic Stress Scale, and 0.895 for the WHOQOL-BREF.

More than half of the students (n = 230, 52.3%) reported better QoL, the majority perceived a positive educational environment (n = 323, 73.4%) and around half of them had academic stress (n = 225, 51.1%). Considering last year's annual examinations percentages of students as a tool for measuring academic performance, more than half of the students (n = 387, 88%) got 60% to 85% marks and were labelled as better performers. No significant difference was observed between clinical and non-clinical years as mentioned in Table 1.

In Table 2, the results of the Chi-square test to check the association between genders among academic stress, educational environment, academic performance and QoL showed that males had significantly more academic stress (p<0.05) as compared to females Those students who perceived a positive educational environment and better QoL had better academic performance (p<0.05). Among males, a significant majority of 55.8% experienced high levels of academic stress, whereas only 35.7% of them demonstrated academic performance in the category of 85 to 100%. In contrast, among females, a majority of 53.7% reported no academic

**Table 1. Basic details of demographic variables (n = 440).**

| S. No | Variables | | n (%) |
|---|---|---|---|
| 1 | Gender | Male | 224(51%) |
| | | Female | 216(49%) |
| 2 | Speciality | MBBS | 336(76.4%) |
| | | BDS | 104(23.6%) |
| 3 | Family Structure | Joint family | 166(37.8%) |
| | | Nuclear family | 274(62.2%) |
| 4 | Year of Study | Pre-clinical | 213 (48.4%) |
| | | Clinical | 227 (51.6%) |
| 5 | Academic Performance | 10% to 35% | 0 |
| | | 35% to 60% | 40(9%) |
| | | 60% to 85% | 387(88%) |
| | | 85% to 100% | 13(3%) |
| 6 | WHOQOL Scale | Better QoL | 230 (52.3%) |
| | | Poor QoL | 210 (47.7%) |
| 7 | DREEM Inventory | Excellent environment | 26 (5.9%) |
| | | More positive than negative environment | 323 (73.4%) |
| | | Plenty of problems | 91 (20.7%) |
| 8 | Academic Stress scale | Higher academic stress | 225 (51.1%) |
| | | No academic stress | 215 (48.9%) |

stress and an even higher proportion of 64.3% achieved academic performance in the top category.

In Table 3, Spearman correlation analysis found a moderate but significant, association between academic performance and stress; better academic performance is associated with lower academic stress, and a significant but moderate and positive association was found with a better QoL. In addition, the results showed a non-significant negative correlation with the educational environment. Whereas the quality of life has a significantly moderate but negative correlation between academic stress and educational environment, but a significantly moderate and positive correlation between the educational environment and academic stress.

## Qualitative results

We used Braun and Clarke's inductive thematic analysis approach to code the transcripts. The transcripts were read in an iterative pattern by the researchers and manually coded. Around

**Table 2. Chi-square test for association between demographic variables (gender & academic performance) with DREEM inventory, WHOQOL and academic stress scale (n = 440).**

| Variables | | Male n(%) | Female n(%) | $X^2$ (P- value) | Academic Performance | | | $X^2$ (p-value) |
|---|---|---|---|---|---|---|---|---|
| | | | | | 35%-60% | 60%-85% | 85%-100% | |
| DREEM Inventory | Excellent | 11 (4.9%) | 15 (6.9%) | 2.33 (.312) | 5 (20.8%) | 19 (5.1%) | 2 (4.8%) | 20.3 (**.002***) |
| | Positive | 161 (71.9%) | 162 (75%) | | 16 (66.7%) | 267 (72.2%) | 38 (90.5%) | |
| | Plenty of Problems | 52 (23.2%) | 39 (18.1%) | | 2 (4.8%) | 38 (90.5%) | 2 (4.8%) | |
| WHOQOL | Better QOL | 112 (50%) | 118 (54.6%) | .945 (.331) | 8 (33.3%) | 190 (51.4%) | 29 (69%) | **9.14 (.027***) |
| | Poor QOL | 112 (50%) | 98 (45.4%) | | 16 (66.7%) | 180 (48.6%) | 13 (31%) | |
| Academic Stress | High Stress | 125 (55.8%) | 100 (46.3%) | **3.98 (.046***) | 11 (45.8%) | 195 (52.7%) | 15 (35.7%) | **8.45 (.038***) |
| | No Stress | 99 (44.2%) | 116 (53.7%) | | 13 (54.2%) | 175 (47.3%) | 27 (64.3%) | |

**Table 3. Spearman coefficient correlation between academic performance, academic stress scale, WHOQOL scale and DREEM inventory (n = 440).**

| Measure | Academic Performance | Academic Stress Scale | WHOQOL Scale | DREEM Inventory |
|---|---|---|---|---|
| **Academic Performance** | 1 | | | |
| **Academic Stress Scale** | -0.084* (0.078) | 1 | | |
| **WHOQOL Scale** | 0.121* (0.011) | -0.097* (0.043) | 1 | |
| **DREEM Inventory** | -0.036 (0.438) | 0.342** (0.000) | -0.284** (0.000) | 1 |

112 codes emerged in the first cycle of coding which converged into 9 categories of codes. The categories with mutually exhaustive meanings were converted into 5 themes.

### Theme 1: Challenging educational environment

Most Healthcare professionals (n = 8, 80%), reported that medical and dental students are often exposed to challenging educational environments. Medical students have to fulfil multiple tasks in a short span of time and sometimes in the emotionally stressful situations of dealing with patients in pain. Such circumstances lead to perceived academic stress and affect the academic performance of the students.

> *Students have to fulfil multiple tasks along with dealing with the patients in a short time period, in which they have to achieve their targets in educational settings. (G = 8).*

However, two interviewees were of the opinion that a challenging educational environment makes the students strong, and they perform well.

> *Challenging Educational Environment is useful in the sense that it makes people strong. (G2)*

> *Challenging Educational Environment leads to better performance under stress. (G7)*

### Theme 2: Psychological needs and support

All the interviewees (n = 10, 100%) thought that early psychological assessment and counselling are very important for students and should be made a core component in the early years of a student's life at college.

> *For every student, it's essential to have an early psychological assessment and access to a counsellor to discuss their problems and stressors and to help them cope with their problems. (G10)*

### Theme 3: Individual differences

The majority of the interviewees (n = 9, 90%) were of the opinion that students differ from each other concerning personality traits and individual characteristics, so they should be facilitated accordingly. The role of mentorship is extremely important for the better mental health performance of the students.

> *Students are different from each other, in every aspect, even having different personality traits that reflect in their appearance and behaviour. They face different sorts of circumstances during their course of studies. A single factor cannot affect all students in the same manner. (G8)*

However, one participant was of the opinion that in this age group, almost all the students are affected similarly as their challenges are somewhat the same.

*Almost all the students are of the same age and face the same challenges, which lead them to academic stress and poor academic performance. (G8)*

## Theme 4: Social and family relationships

Most of the interviewees (n = 8, 80%) believed that social life and relationships are very important for students. A supportive parent and sibling relationship help the students to develop a positive way of thinking leading to a balanced life, less academic stress, and better performance.

*The role of social and family relationships is very important in terms of a better quality of life and academic performance. A strong supportive home environment and family can reduce academic stress. (G8)*

However, two of the healthcare professionals narrated that social relationships have little importance in the life of medical students as they don't have time to maintain them.

*Medical and Dental students are very busy in their everyday routines, so their social relationships affect them little. (G3)*

*Medical and Dental students can't maintain their social life at the expense of their precious time, so social relationships don't affect them significantly. (G9)*

## Theme 5: Adjustment issues

Most of the interviewees (n = 9, 90%) reported that newly admitted students have to face adjustment issues in medical schools and especially in the hostels because of changes in the environment. These early stressors affect the students' quality of life and ultimately their academic performance. Students should be given some window period to adjust to a new environment and the proper guidance from seniors and teachers is very important in helping them to adjust to the new environment.

*Many students face adjustment difficulties at the start of the academic period, because of homesickness, along with roommates, so it's quite difficult for them and it affects their QoL and also their performance. (G9)*

*It should be mandatory and important to give the space, a lively and friendly environment for the students to adjust to this challenging environment. (G10)*

However, one of the interviewees believed that students have already experienced hostel life and long study hours during their entry test preparations and intermediate levels, so the adjustment to a new environment is not a big deal.

*Students have already gone through such situations before, most of them have experienced hostel life and studied for more than 15 hours a day that's why they are in medical college. (G6)*

The quantitative and qualitative analysis results revealed a significant effect of academic stress and educational environment on the academic performance and QoL of medical and dental students. Fig 1 illustrates the sequential explanatory design employed in the study. The

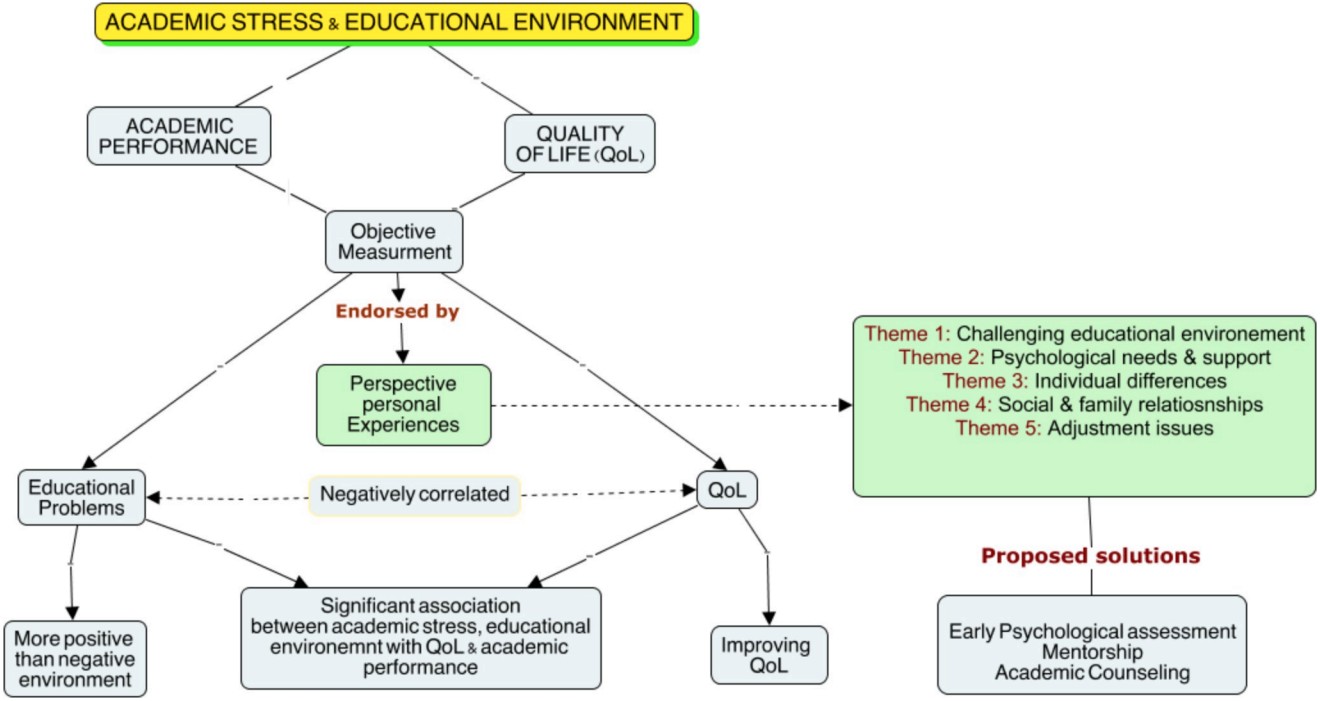

**Fig 1. The sequential explanatory research design.** Qualitative themes enriched the quantitative findings.

qualitative input from healthcare professionals who were the counsellors of the students enriched the interpretation of quantitative findings.

## Discussion

This study is an attempt to draw attention to the effects of the educational environment, and academic stress on the academic performance and QoL of medical and dental students and also gauge the perspectives of healthcare professionals regarding the stress levels of students in a tough educational environment. The reliability of scales, used in this study, is internally consistent and reliable. The reliability of the World Health Organization quality of life scale (WHOQOL) is in line with other studies, [15, 23, 24]. The reliability of the DREEM inventory in our study was slightly less than the reliability of the same tool found in other studies [25].

In our study, the majority of the participants perceived the educational environment as positive (73.4%), which is in alignment with other studies [26–29]. The results of our study regarding the high prevalence of academic stress in medical and dental students have been supported by the study conducted in the USA and at King Saud University, KSA where the prevalence of academic stress among students was 30% and 63%, respectively, affecting the academic performance of the students [1, 2]. In our study, males had more academic stress than females which is contrary to the results of another study, in which there was no significant gender difference [11]. A systematic review identified 23 studies that evaluated the relationship between academic assessment and psychological distress among female medical students which is in line with our research findings [28].

Our study results found no significant differences in opinion among male and female students in terms of DREEM inventory which is in line with few studies that report insignificant gender differences in opinion regarding the educational environment [30–33]. However, these

findings are in contrast with few studies, in which female students perceived the educational environment as more positive than male students [27–29].

The correlation between academic stress and quality of life has been aligned with the findings of Garg K et al, in North India [3]. Whereas a better quality of life leads towards better academic performance and vice versa and the importance of a properly balanced life for better academic performance and better mental health has been supported by many recent studies carried out in different parts of the world [7, 9, 10, 12]. Academic stress was significantly correlated with the educational environment which is in line with the findings of another study [34]. Similarly, a systematic review and a survey carried out at Islamia University Bahawalpur reported a negative association between academic stress and QoL among university students, which is in accordance with our research findings [35, 36].

In our study, students of pre-clinical and clinical years have perceived an equal amount of academic stress which is in contrast to the findings of another study, which found a high prevalence of stress in students of pre-clinical years [11]. Similarly, the findings of a study conducted in Ghana confirmed that the students in senior high schools with a pleasant physical environment perform better which is contrary to our research findings [37].

The association between academic stress and QoL with academic performance in our study has been supported by other studies [10]. Whereas the correlation between academic performance and the educational environment in our study is contradictory to the findings of another study conducted in Pakistan, in which a better perceived educational environment leads to better academic performance [13].

In our study, the majority of the students (74%) perceived the educational environment better and had better QoL (62%) which is contrary to many other studies across the globe in which medical and dental students report average quality of life and perceive the educational environment as negative [2, 3, 7]. The difference may be due to more stress on the professional and ethical obligations of the institutes as described in the vision and mission and also reflects the strong family structure in this particular region. Students' counselling services led by healthcare professionals are also available right from the early years of the academic journey at these schools.

The qualitative phase supplemented the quantitative results by offering deeper insights into the factors influencing academic stress, educational environment, and their impact on academic performance and quality of life among medical and dental students. The most important verbalism of our qualitative part; lack of time, the imbalance between social and academic life, distressful daily situations in college and hospital environments and individual personality differences of students has been supported by many different studies [4, 6, 10]. Extensive courses and training in medical and dental studies lead towards a change in psychological and physical QoL, which has been supported by a few studies [8, 12]. Counselling services at the undergraduate level must be provided to ensure support in personality development.

Our studies showed no significant effect of family structure on students' academic performance which is aligned with the Azumah FD study's results and contradicts Hanul P's study, which shows a significant effect of family structure [38, 39]. Family support and good relationships are very important for better academic performance as per our qualitative result which is similar to a study carried out in Bangladesh [40]. This similarity can be due to the common cultural backgrounds, socioeconomic status, and family structures of the communities of Pakistan and Bangladesh.

A study conducted in nursing college reported that work overload and academic stress lead to a challenging educational environment leading to burnout of students' psychological and mental health, supporting our qualitative theme of a challenging educational environment in the medical career that leads to improvement in academic performance to but causes

psychological distress [41]. Nursing students are integral members of the healthcare sphere and share the same stressful environment as medical institutions and this implies a plausible parallelism in the outcomes. The qualitative findings in our study endorse the challenging educational environment experienced by medical and dental students, impacting academic stress and performance, and are supported by the literature [8, 12, 41]. Psychological support, individual differences, social and family relationships, and adjustment issues in shaping students' experiences and outcomes are the influencing factors well comprehended in the literature [4, 10, 12].

## Conclusion

The study is a significant contribution towards finding a correlation between academic stress, educational environment, QoL, and academic performance in medical and dental students. Our study also showcases the varying findings regarding gender differences in academic stress and perceptions of the educational environment, indicating the need for further research in this area. Additionally, the perception that a pleasant physical environment leads to better academic performance was contradicted by the present research. Individual differences exist in the personality traits of the students, that affect academic performance and add to academic stress. A challenging educational environment leads towards better academic performance but increases academic stress. Early psychological assessment, support and counselling should be made a part of the routine activities of students at college.

## Limitations

The main limitation of our study is that we have taken two institutes only because of the time limit and feasibility of the research. Apart from that our sample size in the quantitative part is a bit small, so in the future will focus on the longitudinal type of research with a large sample size.

## Supporting information

**S1 Data.**
(ZIP)

**S2 Data.**
(BIN)

## Acknowledgments

We are highly grateful to all the students who took a keen interest in giving data. We are grateful to the healthcare professionals and the administration of the private institutes for encouraging, supporting and helping us in the completion of our research.

## Author Contributions

**Conceptualization:** Muhammad Hassan Wahid, Mifrah Rauf Sethi, Neelofar Shaheen, Kashif Javed, Ijlal Aslam Qazi, Muhammad Osama, Abdul Ilah, Tariq Firdos.

**Data curation:** Muhammad Hassan Wahid, Mifrah Rauf Sethi, Neelofar Shaheen, Kashif Javed, Ijlal Aslam Qazi, Muhammad Osama, Abdul Ilah, Tariq Firdos.

**Formal analysis:** Muhammad Hassan Wahid, Mifrah Rauf Sethi, Neelofar Shaheen, Kashif Javed, Ijlal Aslam Qazi, Muhammad Osama, Abdul Ilah, Tariq Firdos.

**Investigation:** Muhammad Hassan Wahid, Mifrah Rauf Sethi, Neelofar Shaheen, Kashif Javed, Ijlal Aslam Qazi, Abdul Ilah.

**Methodology:** Muhammad Hassan Wahid, Mifrah Rauf Sethi, Neelofar Shaheen, Kashif Javed, Ijlal Aslam Qazi, Muhammad Osama, Abdul Ilah, Tariq Firdos.

**Project administration:** Muhammad Hassan Wahid, Mifrah Rauf Sethi, Neelofar Shaheen, Kashif Javed, Muhammad Osama, Abdul Ilah, Tariq Firdos.

**Resources:** Mifrah Rauf Sethi, Kashif Javed, Ijlal Aslam Qazi, Muhammad Osama, Abdul Ilah, Tariq Firdos.

**Software:** Muhammad Hassan Wahid, Mifrah Rauf Sethi, Neelofar Shaheen.

**Supervision:** Mifrah Rauf Sethi.

**Validation:** Muhammad Hassan Wahid, Abdul Ilah.

**Visualization:** Muhammad Hassan Wahid, Mifrah Rauf Sethi, Neelofar Shaheen, Kashif Javed, Ijlal Aslam Qazi, Muhammad Osama, Abdul Ilah, Tariq Firdos.

**Writing – original draft:** Muhammad Hassan Wahid, Mifrah Rauf Sethi, Neelofar Shaheen, Kashif Javed, Ijlal Aslam Qazi, Muhammad Osama, Abdul Ilah, Tariq Firdos.

**Writing – review & editing:** Muhammad Hassan Wahid, Mifrah Rauf Sethi, Neelofar Shaheen, Kashif Javed, Ijlal Aslam Qazi, Muhammad Osama, Abdul Ilah, Tariq Firdos.

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
