## [Decision Letter · Decision Letter 0]

20 Jun 2023

PONE-D-23-12972“Effect of academic stress, educational environment on academic performance & quality of life of medical & dental students; Gauging the understanding of health care professionals on factors affecting stress: A mixed method study”PLOS ONE

Dear Dr. Shaheen,

Thank you for submitting your manuscript to PLOS ONE. After careful consideration, we feel that it has merit but does not fully meet PLOS ONE’s publication criteria as it currently stands. Therefore, we invite you to submit a revised version of the manuscript that addresses the points raised during the review process. Please submit your revised manuscript by Aug 04 2023 11:59PM. If you will need more time than this to complete your revisions, please reply to this message or contact the journal office at plosone@plos.org. Please include the following items when submitting your revised manuscript:A rebuttal letter that responds to each point raised by the academic editor and reviewer(s). You should upload this letter as a separate file labeled 'Response to Reviewers'.A marked-up copy of your manuscript that highlights changes made to the original version. You should upload this as a separate file labeled 'Revised Manuscript with Track Changes'.An unmarked version of your revised paper without tracked changes. You should upload this as a separate file labeled 'Manuscript'.

We look forward to receiving your revised manuscript.

Kind regards,

Mukhtiar Baig, Ph.D.

Academic Editor

PLOS ONE

Journal Requirements:

2. Please ensure that you include a title page within your main document. You should list all authors and all affiliations as per our author instructions and clearly indicate the corresponding author.

3. Please ensure that you refer to Figure 1 in your text as, if accepted, production will need this reference to link the reader to the figure.

Reviewers' comments:

Reviewer's Responses to Questions

**Comments to the Author**

1. Is the manuscript technically sound, and do the data support the conclusions?

Reviewer #1: Partly

Reviewer #2: Partly

Reviewer #3: Partly

Reviewer #4: Yes

2. Has the statistical analysis been performed appropriately and rigorously? 

Reviewer #1: Yes

Reviewer #2: Yes

Reviewer #3: No

Reviewer #4: Yes

3. Have the authors made all data underlying the findings in their manuscript fully available?

Reviewer #1: No

Reviewer #2: No

Reviewer #3: No

Reviewer #4: Yes

4. Is the manuscript presented in an intelligible fashion and written in standard English?

Reviewer #1: No

Reviewer #2: No

Reviewer #3: Yes

Reviewer #4: Yes

5. Review Comments to the Author

Reviewer #1: Thank you for the opportunity to review your submitted paper.

This paper is an analysis of the relationship between academic performance and quality of life, academic stress and academic environment among medical and dental students in Pakistan. I will mention a few points that were difficult to understand and that I think need to be added.

P2L39:The authors wrote" (Phase" , but it is a pair")" is missing.

P2L54:The space is still painted yellow.（Before brackets）

P9Table2:p-value between DREEM and academic performance is .002 that is <0.05, but there is no *.

The subjects of this study were a mix of medical and dental school students. Were there any special characteristics of each school?

The authors utilized a mixed methods study in this paper, but the purpose of the qualitative study in particular is unclear.

Are healthcare professionals the appropriate to consider their perspectives on student's academic stress?

It seems to me that the results from the qualitative study are rarely reflected in the discussion and conclusions.

We would like to see a more complete discussion of the results obtained in this study, as it is limited to mentioning the agreement or disagreement with previous studies.

Reviewer #2: 1. Line 35 - 37: Abstract did not highlight the research gap the study is trying to address.

2. Line 39: Phase?

3. Line 62 103 (Introduction): Introduction describes previous study findings but did not highlight existing problem fuelling the motivation to conduct the present study.

4. Line 71-72: Please include citation.

5. Line 76: Please include citation.

6. Line 93-94: Please include citation.

7. Line 119: Please justify the decision to exclude first year medical students.

8. Line 135: In the result section, the authors reported WHO-QOL score categorically as Better vs. Poor QOL. Please indicate the cut-off point use to determine whether the participants belong to either better or poor QOL.

9. Line 146: Suggestion: It is recommended for authors to indicate what does this range of score suggests e.g., Very low, Low, High, Very high.

10. Line 154-156: Please recheck sentence structure. Please elaborate how the scores were computed or the cut-off score to determine if participants belong to the excellent/ more positive than negative environment/ plenty of problems category.

11. Line 157: Please indicate the cut-off score for no vs. high academic stress.

12. Line 162: Not sure what the phrase "(4 into 8)" means in this sentence.

13. Line 166: Please justify sample size estimation for the qualitative interviews. Who are these healthcare professionals... are they nurses, physicians, dietician?

14. Line 180-182: Please indicate the r coefficient value for weak, moderate or strong correlation.

15. Line 191: Please amend to (N = 440)

16. Line 200 - 202: Please address the academic performance of students who have different level of stress as indicated by Table 2.

17. Line 203 - 209: Instead of highlighting the direction of correlation, i.e., positive/negative, it is suggested for author to directly explain what does the negative/positive correlation means. For example, the sentence "The Pearson correlation showed a significant moderate but negative correlation between academic performance and academic stress and similarly showed a significant moderate but positive correlation with the quality of life." can be revised to "Correlation analysis found moderate, but significant, association between academic performance and stress; better academic performance is associated with lower academic stress".

18. Table 2: Is the symbol "*" indicates significant differences between examined variables? If yes, shouldn't p value for DREEM inventory i.e., p =0.002 mark with "*"?

19. Line 310-369 (Discussion): While the Discussion do address how the findings of current study compared to previous study, it definitely needs more work as it is lacking analytical insight from the authors. Also, the authors did not demonstrate how the findings in Phase 2 explains/relates to Phase 1's quantitative findings.

20. Line 314 - 317: This information should be placed in the Results section.

21. Line 394: In Line 177, the author mentioned that the study was conducted in a private institution in Peshawar, Pakistan. However, the author stated in Line 394 that the study involves two institutes. Please clarify which is the correct information.

22. Figure is not explained in the text.

Reviewer #3: The study assesses the effect of academic and educational environment on academic performance and quality of life of medical and dental students. Few comments for the authors.

General comments

The authors should follow the same sequence throughout the manuscript i.e. quantitative before qualitative in the abstract, results and discussion. The authors should be focused on their aims/objectives and let the results and discussion align with aims/objectives.

Specific comments

Abstract

i. Let your results align with the aim of the study. Especially, the correlation aspect of the study. We are interested in the influence of stress and educational environment on academic performance and QoL.

ii. Since your design is sequential-Exploratory Mixed-Methos as stated in your methods section, can you report your results in that sequence (Quantitative before qualitative).

iii. Can you make the abstracts' results a paragraph?

Introduction

i. Can you add two or more statements on the association/relationship between educational environment and quality of life?

Methodology

i. Can you add a statement or more about the validity of the instruments? Let the readers know how you classified the parameters/variables of the instruments measured.

ii. What informed the sample size of 10 for qualitative phase? Is it adequate?

Data analysis

i. Three of the instruments used were on ordinal level of measurement, Spearman correlation will be appropriate instead of Pearson correlation.

ii.Chi-square test measures association and not mean differences.

Results

i. Under the methodology above, you stated the age range was between 19 and 25 years and students older than 18 were included. Please reconcile why 18 to 25 years were included.

ii. Table 2 showed results of association between the variables. Therefore, the two sentences cannot be said to be true based on table 2 (lines 200-202). If you are interested in mean differences, please run the appropriate test (ManWhiney U test for gender difference of stress, t-test or ANOVA for academic performance).

iii. Please reconstruct results of correlation for clarity. lines 203-209

Discussion

i. Focus on your aims and objectives in the discussion. Since your objective is to assess the effect of academic stress and educational environment on academic performance and QoL, let your discussion align with these. You can expand your discussion along this line especially, the correlation aspect of the study.

ii. Your analysis does not support some of the results' discussion

iii. No mention of years of study influences academic stress in the results section. How come discussing it

iv. Let your conclusion be within the limit of your findings. Don't extrapolate nor draw inferences.

Figure

No reference to the figure in the manuscript.

Reviewer #4: 1. The study presents the results of original research.

2. Results reported have not been published elsewhere.

3. Experiments, statistics, and other analyses are performed to a high technical standard and are described in sufficient detail.

4. Conclusions are presented in an appropriate fashion and are supported by the data.

5. The article is presented in an intelligible fashion and is written in standard English.

6. The research meets all applicable standards for the ethics of experimentation and research integrity.

7. The article adheres to appropriate reporting guidelines and community standards for data availability.

However, the following issues need to be addressed:

- The title is too long and confusing, consider editing it

- Ensure abbreviations are defined in their full meaning at their first use in the abstract and the main article

- Line 144-147: Provide citation/reference for these statements

- Line 172-173: …..“The interview guide was developed by following the protocol for

questionnaire development and was validated by the experts.”…..CITE THIS STATEMENT

- Line 175-182: Cite these statements

- Line 184-188: Cite these statements

6. PLOS authors have the option to publish the peer review history of their article (what does this mean?). If published, this will include your full peer review and any attached files.

Reviewer #1: No

Reviewer #2: No

Reviewer #3: No

Reviewer #4: **Yes: **Fred Ssempijja

<quillbot-extension-portal></quillbot-extension-portal>

---

## [Author Response · Author response to Decision Letter 0]

3 Jul 2023

We would like to express our gratitude to the esteemed reviewers for their reviews and generous consideration in advancing the study further. We have diligently endeavoured to incorporate all the recommendations put forth by the reviewers in our manuscript. Below, you will find a comprehensive response to each of the comments of the reviewers.

Reviewer #1: 

Abstract

Comment: P2L39: The authors wrote" (Phase" , but it is a pair")" is missing.

Response: The study had 2 phases so this has been corrected to clarify and also highlighted in the manuscript (L53)

Comment: P2L54: The space is still painted yellow.（Before brackets）

Response: This mistake has been corrected and highlighted in the manuscript.

Comment: P9 Table2: p-value between DREEM and academic performance is .002 that is <0.05, but there is no *.

Response: The * has been placed and highlighted. (Table 2)

Comment: The subjects of this study were a mix of medical and dental school students. Were there any special characteristics of each school?

Response: These were two sister institutes for MBBS & BDS under one organization. The overall environment in the institutes is similar because similarities in administration and students face similar kind of challenges whether they belong to MBBS or BDS. So it was better to have both institutes together in one study.

Comment: The authors utilized a mixed methods study in this paper, but the purpose of the qualitative study in particular is unclear.

Response: The purpose of the qualitative study is to explore the individual in-depth perspective of experienced faculty (the health care professionals) in detail about all the circumstances they experience during their daily dealings with students regarding academic stress and educational environment. The faculty also highlighted certain factors highlighted by the students in the quantitative phase to elaborate in more detail as it is intended in the sequential exploratory design.

Figure 1 short description has been added to elaborate the key points (L369-371).

Comment: Are healthcare professionals appropriate to consider their perspectives on student's academic stress?

Response: The healthcare professionals were the faculty members in various departments who had experience dealing with the students to help them in coping with stressful situations during their academic journey. The point has been added and highlighted in the manuscript (L57-58).

Comment: It seems to me that the results from the qualitative study are rarely reflected in the discussion and conclusions. We would like to see a more complete discussion of the results obtained in this study, as it is limited to mentioning the agreement or disagreement with previous studies.

Response: 

The discussion section has been addressed to address the comment. The changes have been highlighted in the manuscript.

Reviewer #2: 

Comment: Line 35 - 37: Abstract did not highlight the research gap the study is trying to address.

Response: The introduction of the abstract has been updated to include the research gap. The changes have been highlighted in the manuscript. (L47-51))

Comment: Line 39: Phase?

Response: The study had 2 phases, so this has been corrected to clarify and highlighted in the manuscript (L53-56).

Comment: Line 62 103 (Introduction): Introduction describes previous study findings but did not highlight existing problem fuelling the motivation to conduct the present study.

Response: The introduction has been updated and changes highlighted in the manuscript. 

(L130-139).

Comment: Line 71-72: Please include citation.

Response: The citation has been highlighted (L102)

Comment: Line 76: Please include a citation.

Response: The citation has been highlighted (L107)

Comment: Line 93-94: Please include citation. 

Response: The citation has been highlighted (L126-127)

Comment: Line 119: Please justify the decision to exclude first year medical students.

Response: First-year medical and dental excluded mainly because of two reasons

1. As per the study protocol, the measure of academic performance was taken as scores of previous professional examinations, which was not possible in the case of newly admitted students 

2. As participants were recruited in February 2022 and the first-year students had joined the institutes in January so students did not face academic stress yet.

The justification has also been added briefly in the manuscript and highlighted.

Comment: Line 135: In the result section, the authors reported WHOQOL score categorically as Better vs. Poor QOL. Please indicate the cut-off point use to determine whether the participants belong to either better or poor QOL.

Response: The following line has been added in the instrument section and highlighted in the manuscript

“Higher scores based on the mean average indicate better quality of life and vice versa”.

Comment: Line 146: Suggestion: It is recommended for authors to indicate what does this range of score suggests e.g., Very low, Low, High, Very high.

Response: The amendments have been done as per the suggestion and changes are highlighted in the manuscript.

Comment: Line 154-156: Please recheck sentence structure. Please elaborate how the scores were computed or the cut-off score to determine if participants belong to the excellent/ more positive than negative environment/ plenty of problems category.

Response: The sentence has been restructured for clarity and highlighted in the manuscript 

(L184 -187)

Comment: Line 157: Please indicate the cut-off score for no vs. high academic stress. 

Response: The following line has been rephrased and highlighted in the manuscript

The 40-items academic stress scale was originally developed by Kim (1970). Five-point Likert scale ranging from “no stress i.e., 0” to “extreme stress i.e., 4”.

Comment: Line 162: Not sure what the phrase "(4 into 8)" means in this sentence.

Response: There were 8 items in each of the 5 subscales. Each item is scored from “no stress i.e., 0” to “extreme stress i.e., 4”. So, the maximum possible score was 4x8=32. The sentence has been rephrased to make it easy to comprehend and highlighted in the manuscript.

Comment: Line 166: Please justify sample size estimation for the qualitative interviews. Who are these healthcare professionals... are they nurses, physicians, dietician?

Response: For qualitative research, purposive sampling lets us select the participants who can give a deep insight into the subject matter of concern so even a small sample size is enough as long as their detailed perspectives fulfil the objectives of the research. The sample size is also governed by saturation, i.e., there comes a point when the same information or perspective is repeated by the interviewees at that point, the researchers can stop taking interviews at the point of saturation. Both these rules were followed by the researchers during qualitative data collection.

The healthcare professionals were basically the faculty members from various departments of the institutes under study who had experience as counsellors of the students and could give a deeper insight into various challenges faced by the students during their stay at medical or dental college. 

Comment: Line 180-182: Please indicate the r coefficient value for weak, moderate or strong correlation.

Response: The following line has been added and highlighted in the manuscript. The (rs) is mentioned because the Pearson correlation is replaced by the Spearman correlation as indicated by the comments of the reviewers

“To check the strength of association, the values of (rs) are 0-0.19 as weak, 0.40-0.59 as moderate, 0.6-0.79 as strong and 0.8-1 as very strong correlation”.

Comment: Line 191: Please amend to (N = 440)

Response: The issue has been addressed and highlighted in the manuscript.

Comment: Line 200 - 202: Please address the academic performance of students who have different level of stress as indicated by Table 2.

Response: The issue has been addressed and highlighted in the manuscript.

Comment: Line 203 - 209: Instead of highlighting the direction of correlation, i.e., positive/negative, it is suggested for author to directly explain what does the negative/positive correlation means. For example, the sentence "The Pearson correlation showed a significant moderate but negative correlation between academic performance and academic stress and similarly showed a significant moderate but positive correlation with the quality of life." can be revised to "Correlation analysis found moderate, but significant, association between academic performance and stress; better academic performance is associated with lower academic stress".

Response: The following amendment has been done and highlighted in the manuscript

“The Spearman correlation analysis found a moderate but significant, association between academic performance and stress; better academic performance is associated with lower academic stress, and a significant but moderate and positive association was found with a better quality of life”.

Comment: Table 2: Is the symbol "*" indicates significant differences between examined variables? If yes, shouldn't the p-value for DREEM inventory i.e., p =0.002 mark with "*"?

Response: The p-value for DREEM inventory i.e., p =0.002 is marked with "*" and highlighted in Table 2

Comment: Line 310-369 (Discussion): While the Discussion do address how the findings of current study compared to previous study, it definitely needs more work as it is lacking analytical insight from the authors. Also, the authors did not demonstrate how the findings in Phase 2 explains/relates to Phase 1's quantitative findings.

Response: The discussion section has been updated and changes have been highlighted in the manuscript.

Comment: Line 314 - 317: This information should be placed in the Results section.

Response: This has been addressed and highlighted in the manuscript

Comment: Line 394: In Line 177, the author mentioned that the study was conducted in a private institution in Peshawar, Pakistan. However, the author stated in Line 394 that the study involves two institutes. Please clarify which is the correct information.

Response: The study was conducted in two sister institutes. One is a medical college and the other one is a dental college. The sentences in the relevant sections have been rephrased to clarify. Relevant changes have been highlighted

Comment: Figure is not explained in the text.

Response: The figure has been referenced in the text and the added text has been highlighted 

Reviewer #3:

Comment: The authors should follow the same sequence throughout the manuscript i.e. quantitative before qualitative in the abstract, results and discussion. The authors should be focused on their aims/objectives and let the results and discussion align with aims/objectives.

Response: The sequence has been updated carefully to align the results and the objectives of the study. The changes have been highlighted (L233-256 & 419-430)

Abstract

Comment: Let your results align with the aim of the study. Especially, the correlation aspect of the study. We are interested in the influence of stress and the educational environment on academic performance and QoL.

Response: The results have been aligned with the objectives of the study. The sentences have been rephrased for clarity and easy comprehension (L233-259)

Comment: Since your design is sequential-Exploratory Mixed-Methos as stated in your methods section, can you report your results in that sequence (Quantitative before qualitative).

Response: The sequence has been updated carefully to align the results and the objectives of the study. The changes have been highlighted 

 Comment: Can you make the abstracts' results a paragraph?

Response: The results are merged into one paragraph. The change has been highlighted in the manuscript.(L61-62).

Introduction

Comment: Can you add two or more statements on the association/relationship between the educational environment and quality of life? 

The amendments have been done and changes are highlighted in the manuscript. (L130-134)

Methodology

Comment: Can you add a statement or more about the validity of the instruments? Let the readers know how you classified the parameters/variables of the instruments measured.

Response: The amendments have been done and changes are highlighted in the manuscript.

Comment: What informed the sample size of 10 for qualitative phase? Is it adequate?

Response: In a qualitative study, the primary focus is on obtaining in-depth and rich insights rather than generalizability to a larger population. The concept of data saturation is crucial in qualitative research, indicating the point at which new information or themes no longer emerge from additional data collection or analysis. By reaching data saturation, it implies that the collected information has reached a point of redundancy and that further interviews or data collection would not significantly contribute to the understanding of the research topic.

Hence, in our study, we deliberately chose to have a relatively small sample size of 10 participants. This decision was based on the established principle in qualitative research that a smaller sample can still provide valuable and comprehensive insights into the phenomenon under investigation. By conducting thorough and iterative interviews with these participants, we were able to delve deeply into their perspectives, experiences, and perceptions, allowing us to gain a thorough understanding of the research topic.

Data analysis

Comment: Three of the instruments used were on ordinal level of measurement, Spearman correlation will be appropriate instead of Pearson correlation.

Response: This issue has been addressed and the Spearman correlation has been incorporated. The changes are highlighted in the manuscript. (Table 3 L288-290)

Comment: The chi-square test measures association and not mean differences.

Response: The correction has been done and the change has been highlighted in the manuscript.

Results

Comment: Under the methodology above, you stated the age range was between 19 and 25 years and students older than 18 were included. Please reconcile why 18 to 25 years were included.

Response: This age range corresponds to the typical age range of students pursuing medical or dental education and ensures a homogeneous sample, facing challenges within this specific context. The age range of 18 to 25 years aligns with the nature and focus of the study, which specifically targets medical and dental students.

Comment: Table 2 showed results of association between the variables. Therefore, the two sentences cannot be said to be true based on table 2 (lines 200-202). If you are interested in mean differences, please run the appropriate test (ManWhiney U test for gender difference of stress, t-test or ANOVA for academic performance).

Response: This has been addressed and highlighted in the manuscript (L245-256).

Comment: Please reconstruct results of correlation for clarity. lines 203-209

Response: The amendments have been done and changes are highlighted in the manuscript. (L245-259).

Discussion

Comment: Focus on your aims and objectives in the discussion. Since your objective is to assess the effect of academic stress and educational environment on academic performance and QoL, let your discussion align with these. You can expand your discussion along this line especially, the correlation aspect of the study.

Response: The amendments have been done and changes are highlighted in the manuscript. (L419-423)

Comment: Your analysis does not support some of the results' discussion

Response: The amendments have been done and changes are highlighted in the manuscript.

(L243-256)

Comment: No mention of years of study influences academic stress in the results section. How come discussing it

Response: The statement has been added in the results section. (L243,244) 

Comment: Let your conclusion be within the limit of your findings. Don't extrapolate nor draw inferences.

Response: The amendments have been done and changes are highlighted in the manuscript. (L461-470)

Comment: Figure

No reference to the figure in the manuscript.

Response: The reference to the figure with a brief explanation has been added and highlighted in the manuscript (L369-371)

Reviewer # 4: 

Comment: The title is too long and confusing, consider editing it

Response: Thank you for your kind suggestion. As it is a mixed method study and there are four variables covering major domains and the ethical approval letter also contains the same title changing the title will be a bit challenging at this point in time. However, considering your and other reviewers’ suggestions we have made amendments to the expression of results and discussion to make the text more comprehendible.

Comment: Ensure abbreviations are defined in their full meaning at their first use in the 

 abstract and the main article

Response: This issue has been addressed and changes have been highlighted in the manuscript.

L54,55 & L167,168

Comment: Line 144-147: Provide citation/reference for these statements

Response: Academic performance has varied definitions in the literature depending on the context in which research had been done. For our study, we have operationally taken annual professional exam results as a measure of academic performance. For clarity, the sentence has been rephrased and highlighted in the document. (L175, 176)

Comment: Line 172-173: …..“The interview guide was developed by following the protocol for

questionnaire development and was validated by the experts.”…..CITE THIS STATEMENT

Response: The statement has been cited and highlighted in the manuscript. (L209)

Comment: Line 175-182: Cite these statements

Response: The citations have been done and changes highlighted in the manuscript (L216-225)

Comment: Line 184-188: Cite these statements

Response: The citations have been done and changes highlighted in the manuscript (L227-231)

RESPONSE TO EDITOR

Thank you for giving the opportunity to improve our manuscript.

The corrections (as advised) have been made as under

1. A rebuttal letter has been prepared and uploaded.

2. revised manuscript with track changes has been uploaded

3. An unmarked copy labelled as "manuscript" has been uploaded.

4. The references (in text cittaions) style have been updated. 

5. Fig 1 has been referenced in the updated version of the manuscript. (an updated version of the figure has been uploaded)

6. The title page has been added in the manuscript, clearly indicating the corresponding author

7. A data output set has been uploaded as compressed zip file.

---

## [Decision Letter · Decision Letter 1]

4 Aug 2023

PONE-D-23-12972R1“Effect of academic stress, educational environment on academic performance & quality of life of medical & dental students; Gauging the understanding of health care professionals on factors affecting stress: A mixed method study”PLOS ONE

Dear Dr. Shaheen,

Thank you for submitting your manuscript to PLOS ONE. After careful consideration, we feel that it has merit but does not fully meet PLOS ONE’s publication criteria as it currently stands. Therefore, we invite you to submit a revised version of the manuscript that addresses the points raised during the review process.

We look forward to receiving your revised manuscript.

Kind regards,

Mukhtiar Baig, Ph.D.

Academic Editor

PLOS ONE

Journal Requirements:

Reviewers' comments:

Reviewer's Responses to Questions

**Comments to the Author**

1. If the authors have adequately addressed your comments raised in a previous round of review and you feel that this manuscript is now acceptable for publication, you may indicate that here to bypass the “Comments to the Author” section, enter your conflict of interest statement in the “Confidential to Editor” section, and submit your "Accept" recommendation.

Reviewer #2: (No Response)

Reviewer #3: (No Response)

2. Is the manuscript technically sound, and do the data support the conclusions?

Reviewer #2: Partly

Reviewer #3: Yes

3. Has the statistical analysis been performed appropriately and rigorously? 

Reviewer #2: Yes

Reviewer #3: Yes

4. Have the authors made all data underlying the findings in their manuscript fully available?

Reviewer #2: Yes

Reviewer #3: Yes

5. Is the manuscript presented in an intelligible fashion and written in standard English?

Reviewer #2: No

Reviewer #3: Yes

6. Review Comments to the Author

Reviewer #2: The authors have addressed the previous comments except for the following:

1. There are still no information provided on the scoring method to categorise the following:

a) WHOQOL (Better vs. Poor)

b) DREEM Inventory (Excellent vs. Positive vs. Plenty of problems)

c) Academic stress (High vs. No)

2. Please include justification for sample size estimation of qualitative study in the text.

3. The content of Discussion is limited to how findings of current study compared to literature. Thus, it definitely needs more work as it is lacking analytical insight from the authors. Please demonstrate how the findings in Phase 2 explains/relates to Phase 1's quantitative findings.

4. Language revision is highly recommended.

Reviewer #3: The authors have addressed most of the comments previously highlighted. The manuscript has improved substantially. Few comments yet to be addressed.

Abstract: You have omitted the clear aim(s) in the abstract. You only provided background and the need.

Methods

"Can you add a statement or more about the validity of the instruments? Let the readers know how you classified the parameters/variables of the instruments measured." You have not addressed these comments.

i. Please provide statements about the validity of each instrument you described.

ii. It is important to indicate mean cutoff mark for better QoL in the description of instrument under methods for readers understanding. In your results, you mentioned the percentage of participants who had better QoL.

iii. It important to indicate mean cutoff mark for having academic stress in the description of instrument under methods for readers understanding. In your results, you mentioned the percentage of participants who had academic stress.

7. PLOS authors have the option to publish the peer review history of their article (what does this mean?). If published, this will include your full peer review and any attached files.

Reviewer #2: No

Reviewer #3: No

---

## [Author Response · Author response to Decision Letter 1]

13 Aug 2023

Reviewer #2: 

Comment:1 

The authors have addressed the previous comments except for the following: 

1. There are still no information provided on the scoring method to categorise the following: 

a) WHOQOL (Better vs. Poor)

b) DREEM Inventory (Excellent vs. Positive vs. Plenty of problems)

c) Academic stress (High vs. No) 

Response: The amendments have been done and changes are highlighted in the manuscript.

The following lines have been revised in the manuscript along with improvement in language. 

WHOQOL

LINE 172-178: Cronbach`s alpha was 0.85, 0.83, 0.62, and 0.81, respectively, for the physical, psychological, social, and environmental domains, and 0.92 for the total scale. The level of internal consistency was acceptable to good [15]. Those who scored more than the mean score of 61.97 were considered to have better QoL and those less than 61 had Poor QoL. 

DREEM

LINE 184-193: The 50-item Dundee Ready Education Environment Measure (DREEM) questionnaire assesses the quality of the educational environment, and responses are rated on a five-point Likert scale, spanning from 0, representing strong disagreement, to 4, signifying strong agreement. The comprehensive cumulative score encompasses values from 0 to 200. Within the 50-item scale, 9 items are negatively stated and necessitate subsequent reverse scoring. The evaluation of students' environmental perception was stratified into the following categories: an aggregate score between 0 and 50 indicated a very poor perception of the environment, a range of 51 to 100 indicated plenty of problems, the interval of 101 to 150 signifies a preponderance of positive aspects relative to negative facets, and a score falling within 151 to 200 was characterised as indicative of an excellent environment.

Academic Stress Scale:

LINE 212-215: Those who scored more than the mean score of 67.13 were considered to have high academic stress and those less than the mean score of 67 had no academic stress. Overall, the higher scores on the mean average indicated more academic stress.

Comment:2

Please include a justification for sample size estimation of qualitative study in the text.

Response: Thank you for your comment. Actually, the researchers identified the faculty members who fulfilled the eligibility criteria and communicated with the faculty member before conducting the interviews. After each interview, the transcripts were analysed. Once the researchers identified that no new insights are coming up and the same information is being generated (i.e. data saturation is reached), they stopped further interviews. Researchers reached the data saturation point after 10 interviews, and it was taken as sample size. This technique of determining the sample size is well established in the literature (Hennink M, Kaiser BN. Sample sizes for saturation in qualitative research: A systematic review of empirical tests. Social science & medicine. 2022 Jan 1;292:114523.)

To clarify this point, the following amendments have been done and changes are highlighted in the manuscript.

LINE 221-229: Faculty members who fulfilled the eligibility criteria were approached to consent to one-to-one interviews. The principal investigator conducted in-depth interviews with healthcare professionals in order to comprehensively investigate their viewpoints concerning the determinants contributing to academic stress among students. Each interview session lasted between 40 to 60 minutes, with the interviews continuing until the point of data saturation, whereby no novel insights were forthcoming. The saturation point was attained upon the completion of 10 interviews. The verbatim content of the audio-recorded interviews was transcribed and subsequently subjected to validation from the interviewees.

Comment:3 

The content of Discussion is limited to how findings of current study compared to literature. Thus, it definitely needs more work as it is lacking analytical insight from the authors. Please demonstrate how the findings in Phase 2 explains/relates to Phase 1's quantitative findings. 

Response: The amendments have been done and changes are highlighted in the manuscript.

LINE 456-458: The qualitative phase supplemented the quantitative results by offering deeper insights into the factors influencing academic stress, educational environment, and their impact on academic performance and quality of life among medical and dental students.

LINE 478-483: The qualitative findings in our study endorse the challenging educational environment experienced by medical and dental students, impacting academic stress and performance, and are supported by the literature [8, 12, 42]. Psychological support, individual differences, social and family relationships, and adjustment issues in shaping students' experiences and outcomes are the influencing factors well comprehended in the literature [4,10,12].

Comment:4

Language revision is highly recommended. 

Response: The amendments have been done and changes are highlighted in the manuscript

Language revisions have been done at various places in the document while addressing the comments of worthy reviewers and also while proofreading the document for clarity. The changes have been highlighted in the following lines in the manuscript:

LINE 45-47: Our study aims to ascertain the effect of academic stress and the educational environment on the QoL and academic performance of medical and dental students, encompassing the perspectives of both students and healthcare professionals.

LINE 184-193: The 50-item Dundee Ready Education Environment Measure (DREEM) questionnaire assesses the quality of the educational environment, and responses are rated on a five-point Likert scale, spanning from 0, representing strong disagreement, to 4, signifying strong agreement. The comprehensive cumulative score encompasses values from 0 to 200. Within the 50-item scale, 9 items are negatively stated and necessitate subsequent reverse scoring. The evaluation of students' environmental perception is stratified into the following categories: an aggregate score between 0 and 50 indicates a very poor perception of the environment, a range of 51 to 100 indicates plenty of problems, the interval of 101 to 150 signifies a preponderance of positive aspects relative to negative facets, and a score falling within 151 to 200 is characterised as indicative of an excellent environment. 

LINE 200-203: DREEM is a widely accepted and universally validated tool, demonstrating an internal consistency of 0.86 and a test-retest reliability of 0.595 (p < 0.001). Previous studies utilizing Confirmatory Factor Analysis have affirmed the good model fit for the five-factor structure of DREEM-50 [16,17]. 

LINE 221-229: Faculty members who fulfilled the eligibility criteria were approached to consent to one-to-one interviews. The principal investigator conducted in-depth interviews with healthcare professionals in order to comprehensively investigate their viewpoints concerning the determinants contributing to academic stress among students. Each interview session lasted between 40 to 60 minutes, with the interviews continuing until the point of data saturation, whereby no novel insights were forthcoming. The saturation point was attained upon the completion of 10 interviews. The verbatim content of the audio-recorded interviews was transcribed and subsequently subjected to validation from the interviewees.

LINE 171-178: Cronbach`s alpha was 0.85, 0.83, 0.62, and 0.81, respectively, for the physical, psychological, social, and environmental domains, and 0.92 for the total scale. Previous studies employing Confirmatory Factor Analysis on the WHOQOL-BREF have documented good to excellent reliability properties and better performance on initial validity tests [15].

LINE 256-261: Our study involved 500 students from a medical and a dental college with a participation rate of 80% (n=440). The average age of the participants was 22.24±1.5 years, with an age range of 18 to 25 years. Of the participants, 51% were male (n=224), and 62.3% (n=274) indicated they lived in a nuclear family setup. The internal consistency of the instruments was strong, with a Cronbach's alpha reliability coefficient of 0.877 for the DREEM inventory, 0.939 for the Academic Stress Scale, and 0.895 for the WHOQOL-BREF.

LINE 398-400: Fig 1, illustrates the sequential explanatory design employed in the study. The qualitative input from healthcare professionals who were the counsellors of the students enriched the interpretation of quantitative findings.

LINE 456-458: The qualitative phase supplemented the quantitative results by offering deeper insights into the factors influencing academic stress, educational environment, and their impact on academic performance and quality of life among medical and dental students.

LINE 476-483: Nursing students are integral members of the healthcare sphere and share the same stressful environment as medical institutions and this implies a plausible parallelism in the outcomes. The qualitative findings in our study endorse the challenging educational environment experienced by medical and dental students, impacting academic stress and performance, and are supported by the literature [8, 12, 42]. Psychological support, individual differences, social and family relationships, and adjustment issues in shaping students' experiences and outcomes are the influencing factors well comprehended in the literature [4,10,12].

Reviewer #3: 

The authors have addressed most of the comments previously highlighted. The manuscript has improved substantially. A few comments are yet to be addressed.

Comment:1 

Abstract: You have omitted the clear aim(s) in the abstract. You only provided background and the need. 

Response: The amendments have been done and changes are highlighted in the manuscript. Due to the work count limit of the abstract, the need and aim of the study have been summarized in one sentence as follows:

LINE 45-47: Our study aims to ascertain the effect of academic stress and the educational environment on the QoL and academic performance of medical and dental students, encompassing the perspectives of both students and healthcare professionals. 

Comment:2

Methods "Can you add a statement or more about the validity of the instruments? Let the readers know how you classified the parameters/variables of the instruments measured." You have not addressed these comments.

i. Please provide statements about the validity of each instrument you described.

Response: Following amendments have been made in the manuscript and highlighted

World Health Organization, Quality of Life Scale: (WHOQOL-BREF):

LINE 172-178: Cronbach`s alpha was 0.85, 0.83, 0.62, and 0.81, respectively, for the physical, psychological, social, and environmental domains, and 0.92 for the total scale. Previous studies employing Confirmatory Factor Analysis on the WHOQOL-BREF have documented good to excellent reliability properties and better performance on initial validity tests [15].

DREEM

LINE 200-203: DREEM is a widely accepted and universally validated tool, demonstrating an internal consistency of 0.86 and a test-retest reliability of 0.595 (p < 0.001). Previous studies utilizing Confirmatory Factor Analysis have affirmed the good model fit for the five-factor structure of DREEM-50 [16,17]. 

Academic Stress Scale:

The 40-item academic stress scale validity studies have not been found in the literature; however, internal consistency of the scale has been demonstrated in various studies and our study also.

The following description has been added and highlighted in the literature:

LINE 210-211: The Academic Stress Scale has a good internal consistency score with a Cronbach alpha of 0.70 [18]. 

ii. It is important to indicate mean cutoff mark for better QoL in the description of instrument under methods for readers understanding. In your results, you mentioned the percentage of participants who had better QoL. 

Response: Following amendments have been made in the manuscript and highlighted

LINE 176-178: Those who scored more than the mean score of 61.97 were considered to have better QoL and those less than 61 were categorized as having poor QoL.

iii. It important to indicate mean cutoff mark for having academic stress in the description of instrument under methods for readers understanding. In your results, you mentioned the percentage of participants who had academic stress. 

Response: Following amendments have been made in the manuscript and highlighted

LINE 212-215: Those who scored more than the mean score of 67.13 were considered to have high academic stress and those less than the mean score of 67 had no academic stress. Overall, the higher scores on the mean average indicated more academic stress.

---

## [Editor Report · Decision Letter 2]

16 Aug 2023

“Effect of academic stress, educational environment on academic performance & quality of life of medical & dental students; Gauging the understanding of health care professionals on factors affecting stress: A mixed method study”

PONE-D-23-12972R2

Dear Dr. Shaheen,

We’re pleased to inform you that your manuscript has been judged scientifically suitable for publication and will be formally accepted for publication once it meets all outstanding technical requirements.

Kind regards,

Mukhtiar Baig, Ph.D.

Academic Editor

PLOS ONE

---

## [Editor Report · Acceptance letter]

22 Aug 2023

PONE-D-23-12972R2 

Effect of academic stress, educational environment on academic performance & quality of life of medical & dental students; Gauging the understanding of health care professionals on factors affecting stress: A mixed method study 

Dear Dr. Shaheen:

I'm pleased to inform you that your manuscript has been deemed suitable for publication in PLOS ONE. Congratulations! Your manuscript is now with our production department. 

Kind regards, 

on behalf of

Professor Mukhtiar Baig 

Academic Editor

PLOS ONE